# Fabrication of Formalin-Fixed, Paraffin-Embedded (FFPE) Circulating Tumor Cell (CTC) Block Using a Hydrogel Core-Mediated Method

**DOI:** 10.3390/mi12091128

**Published:** 2021-09-20

**Authors:** Tae Hee Lee, Young Jun Kim, Woo Sun Rou, Hyuk Soo Eun

**Affiliations:** 1Research Institute for Future Medical Science, Chungnam National University Sejong Hospital (CNUSH), 20, Bodeum 7-ro, Sejong 30099, Korea; 2School of Integrative Engineering, Chung-Ang University, Heukseok-dong, Dongjak-gu, Seoul 06974, Korea; yjclassic@kaist.ac.kr; 3Department of Internal Medicine, Chungnam National University Sejong Hospital, 20, Bodeum 7-ro, Sejong 30099, Korea; 4Department of Internal Medicine, Chungnam National University Hospital, 282, Munwha-ro, Jung-gu, Daejeon 35015, Korea; 5Department of Internal Medicine, College of Medicine, Chungnam National University, 266, Munwha-ro, Jung-gu, Daejeon 35015, Korea

**Keywords:** circulating tumor cell (CTC), formalin-fixed, paraffin-embedded (FFPE), hybrid hydrogel, cell block

## Abstract

Circulating tumor cells (CTCs) are extremely low-frequency cells in the bloodstream. As those cells have detached from the primary tumor tissues and it circulates throughout the whole body, they are considered as promising diagnostic biomarkers for clinical application. However, the analysis of CTC is often restricted due to their rarity and heterogeneity, as well as their short-term presence. Here we proposed formalin-fixed, paraffin-embedded (FFPE) CTC block method, in combination manner with the hydrogel core-mediated CTC accumulation and conventional paraffin tissue block preparation. The hydrogel core specifically captures and releases cancer cells with high efficiency with an immunoaffinity manner. An additional shell structure protects the isolated cancer cells during the FFPE CTC block preparation process. The fabricated FFPE CTC block was sectioned and cytopathologically investigated just the same way as the conventional tissue block. Our results demonstrate that rare cells such as CTCs can also be prepared for FFPE cell blocks and shows great promise for cytopathological CTC studies.

## 1. Introduction

Formalin-fixed, paraffin-embedded (FFPE) biological specimens have been widely used to study diseases with cost-effectiveness and long-term native molecular-state preservation ability [1,2,3]. Currently, FFPE-based immunoassay using immunofluorescence (IF) or immunohistochemistry (IHC) is the mainstay of routine cancer diagnosis based on identifying diagnostic and predictive prognostic cancer biomarkers. In particular, immunohistochemical and immunofluorescence analyses of FFPE cell block samples are traditional cytologic diagnostic tools in both pathology and translational research laboratories [4]. However, FFPE-based immunostaining assay have rarely been applied to study rare cells such as circulating tumor cells (CTCs) and disseminated tumor cells (DTCs), due to the limitation of available technologies to isolate the rare cells and assemble them inside the block. Circulating tumor cells (CTCs), defined as cancer cells detached from the primary tumor site, have been considered as one of the proposing biomarkers to suggest information of the primary tumor status and/or metastatic potential [5,6,7]. Since they are in small quantity (less than in order of 10 CTCs in 1 mL of patients’ blood samples) and have low viability after capturing, therefore, preservation for time-consuming analysis or long-term research is often restricted [8,9]. To lessen the loss of samples in smear preparation, it was used with various adjuvant materials such as agar, egg albumin, and plasma-thrombin. 

Many researchers have pointed out that CTCs can also be preserved in formalin fixation and subsequent paraffin embedment, in the same manner as usual tissue samples [10,11,12]. Despite these attributes, the fabrication of FFPE specimens using CTCs is still challenging due to technical difficulties in collecting these types of cells. First of all, most previous CTC isolation methods were not compatible with the conventional FFPE sample preparation method. For example, the CellSearch^®^ system, the first and only FDA-cleared CTC isolation method, is based on immunomagnetic separation [13,14,15]; thus, it is not able to separate the CTCs from the magnetic beads, and those complexes remain in the entire FFPE sample preparation process. On the other hand, microfluidics-based CTC isolation tools, which have been widely proposed due to their simple and convenient manner, are also hard to link to the FFPE sample preparation method [16,17,18]. Since most microfluidics-based methods have drawbacks in retrieving isolated CTCs, it is more suitable for on-chip identification rather than further analysis after retrieval. Second, more importantly, handling those isolated cells such as the solid sample (i.e., tissue) was arduous even if they have been properly accumulated. For these reasons, a proper core matrix for the nucleation of these rare and randomly distributed cells is urgently required. Therefore, aforementioned limitations motivated us to develop a simple and straightforward platform enabling both specific CTC isolation and direct FFPE sample preparation for clinical use. 

In this study, we present a hydrogel core-mediated CTC accumulation method and the subsequent procedure for FFPE CTC block fabrication. The hydrogel core, a hybridized hydrogel of alginate and poly (vinyl alcohol), has enhanced chemical stability compared to the pristine alginate hydrogel. This hybrid hydrogel has been reported as a stable liquid biopsy platform to isolate the CTCs and exosomes [19]; thus, it is more suitable for the FFPE sample preparation process. To accumulate CTCs specifically, anti-EpCAM antibody, a widely used antibody in CTC research, was immobilized on the surface of the hydrogel core. Then, the surface of the hydrogel core was covered by the additional layer (“Shell”) to protect the isolated CTCs and to embody as a pseudo-tissue. As a proof-of-concept study, we utilized cancer cells as model CTCs to fabricate FFPE CTC block. We first evaluated the capture and retrieval efficiency of the hydrogel core through the spiking tests under both phosphate-buffered saline (PBS) and whole blood. Then, for the major evaluation, we confirmed the stability of the present hydrogel core-shell platform during the FFPE sample preparation process, and finally, we verified the feasibility of the fabricated FFPE CTC block via the pathology study. It is a simple, versatile, and cost-effective method to preserve the instantaneous information of CTCs at a certain moment, and it also provides effortless integration with the conventional analysis performed by pathologists. Therefore, our hydrogel core-mediated CTC accumulation method for FFPE specimen preparation offers a feasible way for cytopathological CTC evaluation.

## 2. Materials and Methods

### 2.1. Cancer Cell and Sample Preparation

In order to evaluate the performance of the present concept, we used Huh-7 hepatocellular carcinoma (HCC) cell lines. The cell line was purchased from Korean Cell Line Bank (Seoul, Republic of Korea). The Huh-7 cell line was incubated under general cell culture conditions, however, their media compositions varied depending on type of cells. Typically, cells were cultured in DMEM media containing 1% penicillin, 10% FBS for 3 days at 37 °C with 5% CO_2_ supplied incubator (MCO-170AIC/UVS, Panasonic, Osaka, Japan). Prior to experiments using cancer cells, Huh-7 was trypsinized and detached. The detached cells were repeatedly spiked in 1x PBS buffer before they were ready to use. The CTC model sample containing cancer cells in whole blood was used to evaluate the clinical utility of the present system. For the experiments using human blood samples, whole blood samples were obtained from two non-HCC patients, respectively, with institutional review board (IRB) approval. Non-HCC patients’ samples were collected from Chungnam National University Sejong Hospital (CNUSH). The study was conducted according to the guidelines of the Declaration of Helsinki, and approved by the Institutional Review Board (or Ethics Committee) of Chungnam National University Sejong Hospital, Sejong, Korea (CNUSH-20-11-012). All the human blood samples were maintained in ethylenediaminetetraacetic acid (EDTA) tubes. Samples were processed within 10 h.

### 2.2. Preparation of Anti-EpCAM Antibody-Immobilized Hydrogel Core

The anti-EpCAM antibody-immobilized hydrogel cores were prepared through ionic crosslinking of polymeric materials. Briefly, 5% (*w*/*v*) alginate solution and 5% (*w*/*v*) PVA solution were blended for enhancing both the mechanical and chemical stability of the gel. The mixture was boiled at 85 °C for 1 h with constant stirring, and then cooled down to room temperatures. The remained aggregates or pellets were whisked vigorously using homogenizer. The droplet of the well-mixed hydrogel blend was loaded into a 100 mM of calcium chloride (CaCl_2_) solution using a homemade extruder, followed by 1 h incubation for hardening gel structure. The spherical-shaped cores were washed using deionized water, and they were kept in the buffered solution at room temperature until further use. Afterwards, the anti-EpCAM antibody was immobilized onto the prepared cores following previously reported literatures [19,20]. Before the modification step, the cores were slightly dehydrated and then fully hydrated for stabilization. Then, the cores were incubated with the mixture consisting of 200 mM of 1-ethyl-3-[3-dimethylaminopropyl] carbodiimide (EDC) and 200 mM of N-hydroxysulfosuccinimide (Sulfo-NHS) to activate EDC-NHS coupling. Thereafter, the activated cores were reacted with anti-EpCAM antibodies (1:10, invitrogen; 1B7, Waltham, MA, USA) at 4 °C for 12 h. The unreacted antibody was washed by PBS buffer solution. The anti-EpCAM antibody-immobilized hydrogel cores were immediately used for the CTC isolation.

### 2.3. Hydrogel Core-Mediated CTC Accumulation

The hydrogel core-mediated cancer cell accumulation was conducted to prepare pseudo-tissue for the fabrication of the cancer cell FFPE blocks. We simply dropped the anti-EpCAM antibody-immobilized hydrogel core into the prepared sample containing cancer cells and then incubated them for an hour on the plate stirrer for enhancing interaction between cancer cells and anti-EpCAM antibodies immobilized on the hydrogel core. After incubation, we gently aspirated the remaining sample, took out the hydrogel core containing cancer cells, and then washed them out in PBS buffer to remove non-target cells and other impurities from the hydrogel core. 

### 2.4. Cell Capture, Viability, and Fluorescence Intensity

The cancer cells in 1x PBS after their capture and release from the anti-EpCAM antibodies immobilized hydrogel core was determined using automated Scepter 2.0 cell counter (Millipore, Burlington, MA, USA). First, capture and retrieval efficeincy was evaluated by the spiking test in 1x PBS with Huh-7 (approximately 5.0 × 10^4^ / mL) HCC cell line. After 1 hour-incubation for an hour, we estimated cell capture efficiency as follow:
$$Capture\;Efficiency=\frac{Initial\;Cell\;Count-Final\;Cell\;Count}{Initial\;Cell\;Count}\times 100\left(\%\right)$$


Thereafter, the isolated cancer cells were released from anti-EpCAM antibodies immobilized on the hydrogel core using 0.5M EDTA solution (Invitrogen, Carlsbad, CA, USA), and the retrieval efficiency was calculated as follow:
$$Retrieval\;Efficiency=\frac{Released\;Cell\;Count}{Isolated\;Cell\;Count}\times 100\left(\%\right)$$


For the viability of cells, samples following to retrieval efficiency were centrifuged once more at 1500 rpm at 24 °C for 5 min. The viability of the cancer cells was determined using trypan blue dye exclusion assay. The cells were stained with trypan blue for 10 min, and then were washed by three times with 1x PBS for removal of non-stained impurity. The live cells were unstained and dead cells were stained. The slides were assessed using Ni-E microscopy (Nikon, Iwasaki, Japan). 

Meanwhile, in whole blood test, we diluted 1 mL of whole blood with 1 mL of 1x PBS to prepare 2 mL of diluted whole blood from 5mL human blood. The CellTracker green (Invitrogen, Inc., Carlsbad, CA, USA) staining was conducted before the spiking process to verify capture and release of the cancer cells (model CTCs). The cancer cells were kept in a staining solution, having 10 μM of CellTracker green, for 30 min, and then it was gently rinsed with 1x PBS solution, according to the manufacturer’s instructions. Those dyed cancer cells were obtained by spiking cell lines (~1 × 10^3^ cells) into the diluted whole blood. After the model CTC capture, the hydrogel core was taken out from the followed by three times wash with 1x PBS. The remained sample lysed erythrocyte using erythrocyte lysis (EL) buffer (Qiagen, Inc., Hilden, Germany) to reduce overlapping cells, following to conduct by counting. In brief, 10 mL of EL buffer was added into 2 mL of diluted whole blood and then incubated for 15 min at 4 °C. After incubation, the collected sample was centrifuged at 400× *g* for 10 min at 4 °C, and repeated twice after the discard of supernatants. The cancer cell pellet was resuspended with 1 mL of 1x PBS. The capture efficiency and retieval efficiency calculations were performed based on the cell counts of dyed cells, with the identical definition mentioned above.

In order to confirm capture efficiency using fluorescence intensity, we conducted to stain of immunofluorescence on anti-EpCAM antibodies immobilized on the hydrogel core before and after cell capture. The cells on anti-EpCAM antibodies immobilized on the hydrogel core were fixed with 4% paraformaldehyde (PFA; Invitrogen) for 15 min, and rinsed with 1× PBS for 3 min. The samples were then dipped again in 0.1% Triton X-100 solutions. After 30 min of permeabilization, a PBS solution containing 2% bovine serum albumin (BSA; Sigma, St. Louis, MI, USA) was supplied for another 40 min. Subsequently, cells were stained with a staining solution composed of Alexa fluor 647 anti-human CD326 (1:100 diluted in PBS, Biolegend, San Dieg, CA, USA). After 1 hour of incubation in the room temperature, the staining solution was gently washed with PBS. The absorption and fluorescence intensity were measured with a microplate spectrophotometer (Thermo Varioskan LUX, Thermo Fisher Scientific, Waltham, MA, USA), and the fluorescence was standardized by fluorescence intensity/EX635/EM680, and relative fluorescence was determined by dividing fluorescence intensity. We confirmed the results of arbitrary unit before and after cell capture on anti-EpCAM antibody-immobilized hydrogel core.

### 2.5. The Surface Analysis Using Field Emission Scanning Electron Microscopes (FE-SEM)

The isolated cancer cells on the hydrogel core were also confirmed using FE-SEM. Since hydrogel itself is not compatible with the SEM imaging process in the vacuum chamber, we conducted a serial dehydration process to extract water from this water-adsorbing polymer and to preserve the structure. In fact, the process of SEM specimen preparation is usually similar to the process of FFPE specimen preparation. Therefore, we almost matched the former one (here, Section 2.5) to the latter one (Section 2.8) except for xylene treatment. In other words, the cancer cell-accumulated hydrogel cores were fixed using 4% paraformaldehyde to retain the morphology of the hydrogel and the cancer cells, and then gradual dehydration was conducted using from 70% ethanol to 100% ethanol. Each dehydration step lasted for 5 min at room temperature. Thereafter, the hydrogel core was cut in half, and the flat side was attached to an SEM stub, followed by coating with a 3.0 nm-thick osmium layer. The acceleration voltage was decided 2.0 kV at a working distance of 8.0 mm to minimize sample damage and charging effects of biomolecules and hydrogels. 

### 2.6. Hydrogel Shell Formation

After capturing cancer cells, the additional hydrogel layer (“shell”) was formed outside the hydrogel core. The composition of this hydrogel layer was identical to the hydrogel core: blending of alginate and PVA. The purpose of this process is to protect the isolated cancer cells from the following procedures for FFPE sample preparation. Briefly, the cancer cell-accumulated hydrogel cores were fully immersed into the viscous hydrogel solution in a very short time (<10 s), and then transferred to a 100 mM of calcium chloride (CaCl_2_) solution one by one. Thereafter, 10-min of further incubation was conducted for hardening gel structure. We denoted them as hydrogel core-shell to distinguish them from the previous hydrogel core. Finally, these pseudo-tissues were washed using deionized water, and they were kept in the buffered solution at room temperature until further use. 

### 2.7. Feasibility Test Using Whole-Body Fluorescent Imaging

The affinity between hydrogel core and cancer cells was verified using the CTC model sample containing fluorescent-labeled cancer cells. The cancer cells (~1.0 × 10^3^) were labeled with CellTracker green, and the details are described to the description in the Section 2.4. After layering with the additional layer, the fluorescence intensity was then monitored using the whole-body fluorescent imaging system (Xenogen In-Vivo Imaging System (PerkinElmer, Waltham, MA, USA). In addition, we obtained the serial images during the FFPE process to prove the possibility of cell loss. Because the fluorescence in-vivo imaging system enables us to analyze hydrogel in a non-invasive manner, we can roughly estimate the amount of the model CTCs placed between the hydrogel core and hydrogel shell. The FITC was detected at wavelengths of 495 nm/519 nm (excitation/emission). Fluorescence images were obtained during an exposure time of 3–5 s (f/stop =  2), and bright-field photographs were also obtained for each imaging time. These images were merged and analyzed using Living Image 4.52 software (Caliper Life Sciences, Hopkinton, MA, USA).

### 2.8. FFPE CTC Block Formation and Hematoxylin & Eosin (H&E) Stain

After cancer cell accumulation, hydrogel cores were covered with an outer hydrogel layer (“Shell”) to prevent the captured cells from cell loss. Then, the hydrogel cores were applied to standard procedures for FFPE tissue block [3,4]. First, the hydrogel cores were cautiously fixed and dehydrated in the order as follows: 4% paraformaldehyde, 70% ethanol, 80% ethanol, 90% ethanol, 95% ethanol, and 100% ethanol. Then, the dehydrated hydrogel cores were incubated with xylene, an intermediate solvent, to infiltrate them into paraffin wax. This procedure was repeated twice for 20 min. Then, the hydrogel cores were placed in embedding cassettes, and the cassettes were immersed into the liquefied paraffin wax (55–60 °C). The wax is solidified around 4 °C; thus, the hydrogel cores containing the isolated cancer cells were embedded in the solidified paraffin wax, similar to the tissues in the common procedure of paraffin tissue block. The prepared blocks were stored for further verification. The CTC FFPE blocks were sectioned for H&E staining. Before CTC FFPE block sections, we placed them at −20 °C, in order to cut them more improved. Sections of 4–5 µm thickness were placed on glass slides, heated at 60 °C for 30 min, and then deparaffinized with xylene and ethanol. The specimens were stained with H&E staining. In brief, the slides were incubated on Mayer’s hematoxylin (ab220365, Abcam) for 3 min and Eosin Y alcoholic solution (BBC Biochemical) for 40 s. Thereafter, the slides were mounted using neo-mount (Merck Millipore, Burlington, MA, USA) and assessed using Ni-E microscopy (Nikon, Iwasaki, Japan). 

## 3. Results

### 3.1. Hydrogel Core-Medicated Cancer Cell Accumulation

As illustrated in Figure 1, the hydrogel core, which hybridized alginate and poly(vinyl alcohol), isolated cancer cells and was then prepared as a pseudo-tissue via additional layering. Briefly, anti-EpCAM antibody-immobilized onto hydrogel cores were utilized to accumulate cancer cells from the sample, and then they were covered again using an identical component of hybrid hydrogel layer to protect the cancer cells from the physical or chemical attacks during the FFPE sample preparation process. In order to designate each sample in the different steps, we used the terms as follows: hydrogel core (without post-layering) and hydrogel core-shell (after post-layering). According to our experience in the preliminary test, the model CTCs were not preserved during the FFPE sample preparation process without the external layer, cause by FFPE fabrication process. The diameter of the hydrogel core was estimated to be 2.53 ± 0.15 mm (*n* = 10), and this value slightly increased to 2.85 ± 0.42 mm (*n* = 10) after outer shell formation. The size of this hydrogel platform was determined for simple and effortless handling during the FFPE sample preparation process. In addition, its millimeter-level size is advantageous in the sectioning of the paraffin block because the location of the hydrogel platform can be confirmed by the naked eye.

### 3.2. CTC Isolation Process with Presenting Superior Capturing Ability and Cellular Viability

In order to evaluate the capture ability of anti-EpCAM antibody-immobilized hydrogel core, we conducted cell concentration before and after cell capture by Scepter cell counter and fluorescence meter in Figure 2. Figure 2a,b calculated in terms of concentration, cancer cells showed variable size components. For the quantitative analysis of cell capture ratio, fluorescence intensity arbitrary unit of unincubated cell from anti-EpCAM antibody-immobilized onto hydrogel was 0.27 ± 0.01. Subsequently, fluorescence intensity arbitrary unit of incubated cell from anti-EpCAM antibody-immobilized onto hydrogel was 0.39 ± 0.09 in Figure 2c. The fluorescence intensity of the incubated cell anti-EpCAM antibody-immobilized onto hydrogel is 1.43 folds higher than that of unincubated cell. This result predicts fluorescent-labeled cancer cells are specifically reacted with anti-EpCAM antibody-immobilized onto hydrogel. In order to demonstrate that the cells are still viable even before and after the anti-EpCAM antibody-immobilized onto hydrogel, we checked the viability of captured cancer cells. 

The cell capture efficiency of the present hydrogel core-based cancer cell accumulation was evaluated using Huh-7 cell line. As shown in Figure 3, the cell capture efficiency of Huh-7 was 92.9 ± 0.9%. This result indicates that the target cancer cells were specifically isolated by anti-EpCAM antibody-immobilized hydrogel core. In addition, we confirmed the retrieval efficiency and viability of the isolated cancer cells. Thanks to the degradable property of the alginate-based hydrogel core, it was able to retrieve the isolated cancer cells effortlessly. We have confirmed that 91.7 ± 10.4% of the isolated cancer cells were released from the hydrogel core, and 89.5 ± 3.5% of them remained still alive. Both retrieval efficiency and viability were comparable to the best performance in the previously announced viable CTC selection methods [21,22]. Additionally, we conducted the efficiency of capture and retrieval of the isolated cancer cells in 2 mL of diluted whole blood which consisted of 1 mL PBS mixed with 1 mL whole blood. The efficiency of capture and retrieval in diluted whole blood was 79.0 ± 3.6% and 86.6 ± 4.9%. In spite of the fact that there was no need to retrieve the isolated cancer cells again in the present concept of our FFPE CTC block, this high retrieval efficiency accompanied with high viability implied the possibility of transferring non-damaged, stress-free cancer cells, which maintained biological information and also reflected original tumor, to the sample FFPE CTC preparation steps.

### 3.3. Feasibility Study of FFPE CTC Block Preparation

Basically, the FFPE sample preparation process involves serial incubation with the chemical fixative and various organic solvents, including ethanol and xylene. Since these processes induced the extensive dehydration of the specimen, hardening the tissues and membrane, the hydrogel core-shell also became gradually denser and smaller as the step proceeded. Considering the definition of hydrogel, three-dimensional water-absorbing polymeric networks, de-swelling accompanied with dehydration was a natural outcome. Unlike the real tissue specimen, this artificial matrix only consists of hydrophilic chains does not have a structural component; thus, its physical dimension is drastically changed along with the level of water uptake. Because the model CTCs (fluorescent-labeled cancer cells) were placed between the hydrogel core and external hydrogel shell, we conducted the feasibility test using whole-body fluorescence imaging, which reads total fluorescence intensity from the entire hydrogel at the target wavelength. By favor of this non-destructive way, we were able to effortlessly quantify the amount of these coagulated cells between core and shell. 

Figure 4a shows the FE-SEM images of the surface of the hydrogel core after 4% PFA and 95% ethanol treatment. On the surface of the hydrogel core, wrinkled surface patterns were formed due to the ionotropic gelation; however, the wrinkles were relatively flat and featureless compared to the usual surface images of ionotropic hydrogels. This might be a result of the serial dehydration and subsequent de-swelling. As shown in Figure 4b, the isolated model CTCs were found on the surface with a substantially shrunken appearance. The diameter of the cells was measured to be around 2 µm. In spite of the fact that these results were obtained from the hydrogel core without the additional layering, those images provide information regarding how the FFPE preparation affects the structure of the hydrogel and cancer cells. 

Figure 4c,d shows the changes in fluorescence intensity of hydrogel core-shell during the process, including PBS, 4% PFA, 95% ethanol, xylene, and paraffin wax, in chronological order. We considered the fluorescence intensity from the control hydrogel core-shell (0.22 ± 0.04 × 10^9^ p^1^∙s^−1^∙cm^−2^) as a background threshold. The measurement was repeated five times and averaged (*n* = 5). After the incubation with the CTC model sample, the hydrogel core-shell showed 2.37 ± 0.43 × 10^9^ p^1^∙s^−1^∙cm^−2^ of fluorescence intensity after 10-min of incubation in PBS. Supposing that there was no loss in PBS solution, the fluorescence intensity of anti-EpCAM antibody-immobilized hydrogel core-shell is approximately 10.6 times higher than that of control hydrogel core-shell. This result indicates the specific affinity to target cancer cells. In the cross-sectional analysis, the fluorescent signals seem to be maintained or slightly enhanced during the FFPE process (Figure 4c); however, total fluorescence intensity per unit volume gradually decreased along with the step by step, although the standard deviations were largely overlapped (Figure 4d). After 30-min of incubation in 4% PFA, hydrogel core-shell showed 2.07 ± 0.38 × 10^9^ p^1^∙s^−1^∙cm^−2^ of fluorescence intensity. It is equivalent to 12.7% of signal decrement. We assume that it is related to the simple extraction of non-bound dye molecules through the dehydration process and the hardening of both hydrogel structure and cancer cells by PFA. Afterwards, the hydrogel core-shell showed 1.82 ± 0.31 × 10^9^ p^1^∙s^−1^∙cm^−2^ and 1.74 ± 0.22 × 10^9^ p^1^∙s^−1^∙cm^−2^ of fluorescence intensity after next two steps (70–95% ethanol and xylene). Additionally, approximately 11.2% of signal decrement was further detected after paraffin embedment; however, this could be caused by attenuation due to the thick and dense paraffin coating. Therefore, the actual signal decrement of core-shell hydrogel during the FFPE preparation process was estimated to be about 26.4%. In other words, more than 70% of the initial fluorescence signal was preserved during this harsh environment.

The cell block containing cancer cell-anti-EpCAM antibody-immobilized onto hydrogel for CTC isolation and H&E staining in Figure 5. We examined the overall cell morphology and the nuclear-cytoplasmic ratio (NC ratio). As showed in Figure 5c, we confirmed the cluster-liked CTC. As in previous work, clustered CTC show a higher ability to form distant metastases than single CTC [5,23], and our results indicate that cluster-liked CTC are applicable to our proposed hydrogel core-based approach. This result implies the advantages of the present method compared to the previous CTC isolation method. For example, the cell separation method using the magnetic-activated cell sorting (MACS) method is also a method that can separate cells while reducing contamination of cells. However, when using MACS, in order to separate various cells at the same time, it is necessary to use specific antibodies and magnetic beads together with magnetic separating tools; it is time-consuming, expensive, and non-applicable to the FFPE block preparation process. Further investigation with a large-scale patients’ sample will be required to evaluate the potential of the present FFPE CTC block preparation method, but we believe it may contribute to connecting laboratory-level CTC isolation techniques to the cytopathological CTC evaluation.

## 4. Conclusions

In this study, we proposed a hydrogel core-mediated CTC accumulation method and the subsequent procedure for FFPE CTC block fabrication. Thanks to the hydrogel-based approach with an immunoaffinity manner and core-shell structure for the protection of the isolated cancer cells, the present platform can be directly utilized to FFPE CTC block preparation; thus, a small can be manipulated similar to the tissue for FFPE-based cytopathological study. The capture and release efficiency in whole blood were 79.0 ± 3.6% and 86.6 ± 4.9%, respectively, and the isolated cancer cells were preserved during the FFPE sample preparation process. Finally, we fabricated the FFPE CTC block with the model samples, conducted an evaluation, and found cluster-like cancer cells. This approach may facilitate the further and deeper investigation of CTCs through the conventional cytopathological method using cell block, thereby revealing their diagnostic and prognostic meaning as a biomarker.

## Figures and Tables

**Figure 1 micromachines-12-01128-f001:**
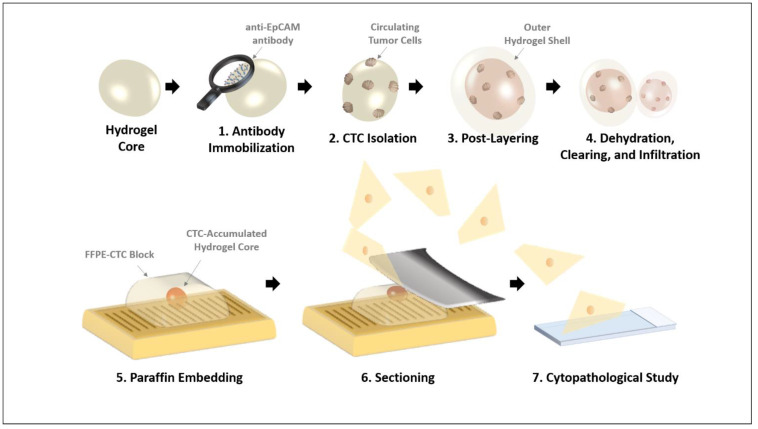
The schematic illustration of formalin-fixed paraffin-embedded (FFPE) CTC block preparation using a hydrogel core-mediated method.

**Figure 2 micromachines-12-01128-f002:**
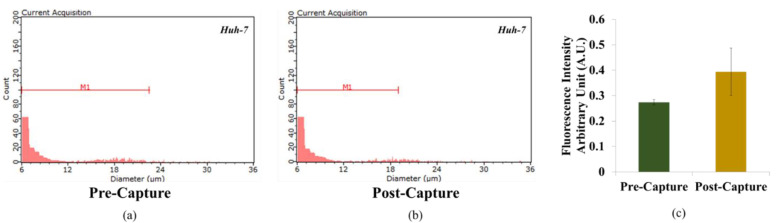
CTC isolation by scepter (**a**) pre and (**b**) post. The capture efficiency of Huh-7 cell line by (**c**) fluorescence intensity.

**Figure 3 micromachines-12-01128-f003:**
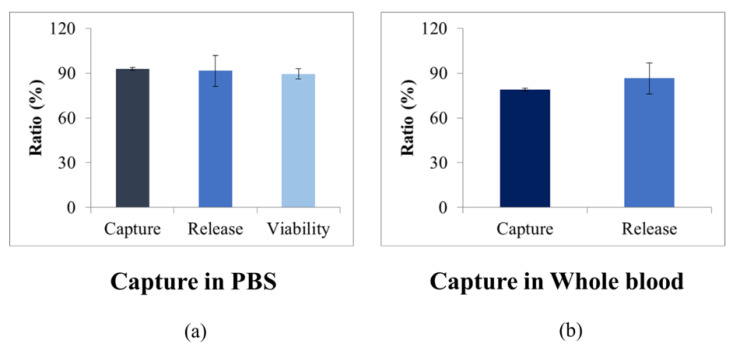
The performance of the anti-EpCAM antibody-immobilized hydrogel core: for ratio of capture, release, and viability from PBS spiked with cancer cell (**a**); for ratio of capture and release from whole blood spiked with cancer cell (**b**).

**Figure 4 micromachines-12-01128-f004:**
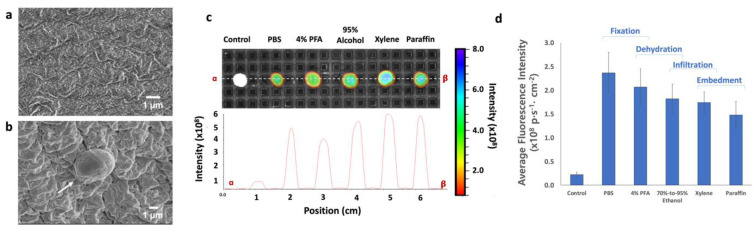
The feasibility test for the present hydrogel core-mediated method. (**a**,**b**) The FE-SEM images of the present hydrogel cores: (**a**) before cancer cell accumulation; (**b**) after cancer cell accumulation. All images were magnified by a factor of 10 K and 6.5 K with an acceleration voltage of 2 kV; (**c**) non-invasive fluorescence tracking image and cross-section view during FFPE sample preparation process (control, PBS-incubated, 4% PFA-incubated, 70–95% ethanol-incubated, xylene-incubated, paraffin-embedded hydrogel core-shell), using whole-body fluorescent imaging system; (**d**) the total fluorescence signal per unit volume during FFPE sample preparation process.

**Figure 5 micromachines-12-01128-f005:**
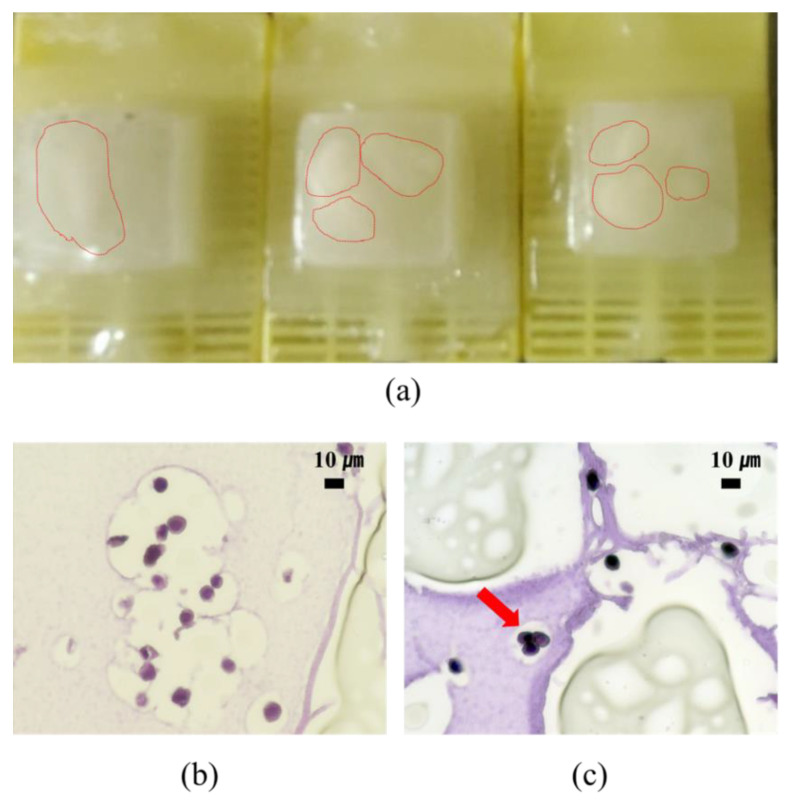
The formalin-fixed, paraffin embedded (FFPE) CTC block: (**a**) the fabricated FFPE CTC block; (**b**) hematoxylin and eosin staining of circulating tumor cells; and (**c**) cluster-liked CTC.

## Data Availability

All data generated from this study are included in this published article and supporting information. Raw data are available from the corresponding author on reasonable request.

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
