# Peer review of "Fabrication of Formalin-Fixed, Paraffin-Embedded (FFPE) Circulating Tumor Cell (CTC) Block Using a Hydrogel Core-Mediated Method"

_micromachines, 2021, doi:10.3390/mi12091128_

Round 1
Reviewer 1 Report
Authors assessed the efficacy of the hydrogel core-mediated CTC accumulation suing formalin-fixed, paraffin-embedded (FFPE) CTC block method. There are so several methods and results, it is difficult to understand authors’ intention. This is because the luck of revealing primary evaluation item.
The CTC accumulation, authors argued, is revealed but the accumulated cells are only cell-line cells not CTC. Using whole blood as medium permit the cell-line cell to be called “CTC”
There are several criticisms listed below.
- The paraffin-embedded method never provide total cell count without completely total thine-section slice of the block, that is not practical.
- Clinically, clustered CTC is crucial. How about this method for clustered CTC?
- Figure 2. It is impossible to identify CTC. Marks such as allow pointing out the CTC is recommended.
- Whole blood incubation is an arm, should be added.
- The cell in figure 5-c is too small to judge the cell to be cancer cell. The photo of higher magnification is needed to show the feasibility of the method.
Author Response
Dear, Reviewer,
The authors thank the referee for the comment. We tried to reflect on your concerns as much as we could. We apologize for confusing the reviewer.
Our research can be divided into two main categories. First : CTC isolation based on anti-EpCAM Antibody-Immobilized Hydrogel Core. Second: Formalin-Fixed, Paraffin-Embedded (FFPE) application using CTC isolation based on anti-EpCAM Antibody-Immobilized Hydrogel Core. In order to explain better, we changed the Figure 2, 3, and 4. The changed Figure 2 and Figure 3 investigate CTC isolation based on anti-EpCAM Antibody-Immobilized Hydrogel Core, before paraffin-embedded method. The changed Figure 4 introduce Formalin-Fixed, Paraffin-Embedded (FFPE) application using CTC isolation based on anti-EpCAM Antibody-Immobilized Hydrogel Core.
Again, apologize for confusing and thank to reviewer very much for your kindly review and constructive comments.
Review1.1 : The paraffin-embedded method never provide total cell count without completely total thine-section slice of the block, that is not practical.
: The authors thank the referee for the comment. We apologize for confusing the reviewer. The cancer cells after their capture and release from the beads was determined using automated Scepter 2.0 cell counter (Millipore). Although we did not count based on paraffin-embedded method, we conducted completely total thine-section slice of the block. The ratio of capture, viability, and release was conducted before paraffin-embedded method. In other words, we measured the ratio of capture, viability, and release based on anti-EpCAM Antibody-Immobilized Hydrogel Core. As the referee mentioned, we also agree that ratio of capture, viability, and release is not practical, after process of paraffin-embedded method.
Review1.2 : Clinically, clustered CTC is crucial. How about this method for clustered CTC?
: The authors thank the referee for the comment. As previous work*, we also recognized “clustered CTC is crucial”. As reviewer comments, we tried to reflect the importance of the clustered CTC in our manuscript. As showed in Figure 5c, we added in results 3.3 Feasibility study of FFPE CTC block preparation “cluster-liked CTC”, which look similar in clustered CTC using Huh-7 cell line. We expect it to be applicable.
* As preview work : Analysis of a Real-World Cohort of Metastatic Breast Cancer Patients Shows Circulating Tumor Cell Clusters (CTC-clusters) as Predictors of Patient Outcomes
Review1.3 : Figure 2. It is impossible to identify CTC. Marks such as allow pointing out the CTC is recommended.
: The authors thank the referee for the comment. We apologize for confusing the reviewer. We thought the identification of cancer cells in the FE-SEM image was possible enough because there was a difference in figure (b) compared to the pristine wrinkle-patterned surface. However, as the reviewers pointed out, sometimes it is hard to recognize the isolated targets (cancer cells) due to the wrinkled surface of the hydrogel. It is also the reason why we had troubles during the SEM imaging process. Therefore, we added the arrow pointing in the image to designate the isolated cancer cells.
Review1.4 : Whole blood incubation is an arm, should be added.
: The authors thank the referee for the comment. We revised the manuscript and conducted additional whole blood test. For the experiments using human blood samples, whole blood samples were obtained with the institutional review board (IRB) approval. For capture of CTCs using on anti-EpCAM Antibody-Immobilized Hydrogel Core, we conducted the 2 mL of diluted whole blood which consisted of 1 mL PBS mixed with 1 mL whole blood obtained from 5 mL human blood. We conducted by spiking cell lines into 2 mL of diluted whole blood. The spiking cell lines were kept in staining solution, having 10 μM of CellTracker green, for 30 min. The efficiency of capture and retrieval in diluted whole blood was 79.0 ± 3.6 % and 86.6 ± 4.9 %. We tried to describe the method of analyses as much as possible in detail through the part of Methods and Results.
Review1.5 : The cell in figure 5-c is too small to judge the cell to be cancer cell. The photo of higher magnification is needed to show the feasibility of the method.
: The authors thank the referee for the comment. We apologize for confusing the reviewer. We tried to reflect about photo of higher magnification. We changed of figure 5c for magnification and enhanced feasibility of the method.

Reviewer 2 Report
See attached file.

Author Response
Dear, Reviewer,
The authors thank the referee for the comment. We tried to reflect on your concerns as much as we could. We apologize for confusing the reviewer.
Our research can be divided into two main categories. First : CTC isolation based on anti-EpCAM Antibody-Immobilized Hydrogel Core. Second: Formalin-Fixed, Paraffin-Embedded (FFPE) application using CTC isolation based on anti-EpCAM Antibody-Immobilized Hydrogel Core. In order to explain better, we changed the Figure 2, 3, and 4. The changed Figure 2 and Figure 3 investigate CTC isolation based on anti-EpCAM Antibody-Immobilized Hydrogel Core, before paraffin-embedded method. The changed Figure 4 introduce Formalin-Fixed, Paraffin-Embedded (FFPE) application using CTC isolation based on anti-EpCAM Antibody-Immobilized Hydrogel Core.
Again, apologize for confusing and thank to reviewer very much for your kindly review and constructive comments.
Review 2
Review2.1 : First of all, the word "circulating tumor cell" should not be mainly used in this manuscript. Even though potential target is CTC, authors are not using actual blood sample, and a model cancer cell line was only used at pure condition without any mixture with other cell lines. The main claim of this manuscript should be the development of a method to observe section of non-adherent cells as similar manner as general tissue section. Title of this manuscript must be corrected and following the correction, contents of the main text should also be reviewed extensively
: The authors thank the referee for the comment. The authors agree that we misused the terminology. We apologize for cancer cell line was only used at pure condition without any mixture with other cell lines. We revised the manuscript and conducted additional whole blood test. For the experiments using human blood samples, whole blood samples were obtained with the institutional review board (IRB) approval. For capture of CTCs using on anti-EpCAM Antibody-Immobilized Hydrogel Core, we conducted the 2 mL of diluted whole blood which consisted of 1 mL PBS mixed with 1 mL whole blood obtained from 5 mL human blood. We conducted by spiking cell lines into 2 mL of diluted whole blood. The spiking cell lines were kept in staining solution, having 10 μM of CellTracker green, for 30 min. The efficiency of capture and retrieval in diluted whole blood was 79.0 ± 3.6 % and 86.6 ± 4.9 %. We tried to describe the method of analyses as much as possible in detail through the part of Methods and Results.
Review2.2 : Next, advantages of the suggested method should be discussed more. There are various methods for concentration and purification of CTCs from blood samples, and various methods for the fabrication of cell section of non-adherent cells are also suggested. Authors should discuss advantages of the suggested method in comparison with previous approaches.
: The authors thank the referee for the comment. As you pointed out, we tried to describe the method of analysis as much as possible in detail through the part of Introduction and Materials & Methods. For example, the first and only FDA-cleared CTC isolation method, the CellSearch® system, utilizes magnetic beads, which are not separated from the isolated CTCs. Therefore, it is not easy to apply to the FFPE preparation procedure. Likewise, microfluidics-based CTC isolation also has drawbacks in the retrieval of isolated CTCs; thus, it is usually impossible to collect the isolated CTCs to fabricate cell pellets for the FFPE preparation procedure. On the other hand, the present hydrogel core-based approach offers high retrieval efficiency without contaminating cancer cells; thus, it offers a simple, versatile, and cost-effective approach to integrate with FFPE sample preparation method.
Review2.3 : S1: L53 "First of all, ..." Authors should indicate objective evidences with suitable references.
: The authors thank the referee for the comment. We revised the paragraph and also added suitable references. Most CTC isolation methods are not compatible with FFPE cell block because they are usually specialized in enumeration; thus, there was no specific way to retrieve the isolated CTCs.
Review2.4 : S2.2: There is not any description about the size of hydrogel particles. At least, average sizes of hydrogels should be described.
: The authors thank the referee for the comment. We revised the manuscript and added a description of the size of the hydrogel core. The average size (diameter)of the hydrogel core was estimated to be 2.53 ± 0.15 mm (n=10). It is slightly increased to 2.85 ± 0.42 mm (n=10) after outer shell formation.
Review2.5 : S2.4: L106 Please explain detailed methods about the formation of "shell" layer.
: The authors thank the referee for the comment. We revised the manuscript and added the materials and methods section (2.6). Briefly, the shell formation process was proceeded by simple dipping of the cancer cell-accumulated hydrogel cores. Then, those hydrogel cores were transferred to the crosslinking solution (CaCl2), followed by further incubation.
Review2.6 : S3.1: Fig. 2b Cells should be indicated with arrow heads. Even the cells were dehydrated, 2-µm diameter is too 2 small as cell sizes. Authors should discuss about too small cell sizes including elimination of the possibility of some artifacts.
: The authors thank the referee for the comment. We agree that the 2-µm diameter is too small for cell sizes. We do not know the exact cause of this massive shrinking more than we expected; but, we assume that the absence of the additional layer (“shell”) in the samples for FE-SEM investigation might be one of the reasons. We conducted the investigation and found another isolated cancer cell on the hydrogel core. Its size is estimated to be 5 µm in diameter. Therefore, We concluded that the description of the specific size could deliver misunderstanding; so, we revised the description and replaced the image with better identification of the isolated cancer cells.
Review2.7 : S3.2: L200, Fig. 2c Unit "p/s/cm2 " might be not appropriate. Using representations such as "p s-1 cm-2 " might get rid of misunderstandings. Authors should describe frequencies of assays (how many time the assay repeated and averaged).
How were the cells labeled with fluorescent dye?
: The authors thank the referee for the comment. We revised the unit (p1∙s-1∙cm-2) to avoid misunderstanding. Also, we added the frequencies of assays and descriptions for the fluorescent labeling. Briefly, the assay was repeated five times with 103 fluorescent-labeled cancer cells and averaged the result.
Review2.8 : The method should be described. Authors should discuss clearly about the cause of fluorescent increase/decrease. For example, the fluorescence decreased 13.8% after the treatment of PFA; the reason should be discussed. In addition, the fluorescence recovered/increased by following steps. This reviewer cannot understand whether the increase of fluorescence is within the error or not, however, possible reasons should be discussed.
: The authors thank the referee for the comment. We added more description regarding the fluorescence signal decrement during the FFPE sample preparation process. Also, we added more data to provide the objective evidence; so, some average values are now slightly revised. For example, 13.8% (it is now calculated to 12.7%) after the treatment of PFA might be related to the simple extraction of non-bound dye molecules through the dehydration process and the hardening of hydrogel structure and cancer cells by PFA. To avoid the misunderstanding, we added Figure 4d, the total fluorescence signal per unit volume. In conclusion, the total fluorescence signal had decreased along with the steps (Figure 4d), unlike the cross-sectional image (Figure 4c).
Review2.9 : S.3.3: Fig. 3 Experimental situation is not clear. Did the authors count cells either in suspension or on gel? What is the conclusion (i.e., new findings) of Figs. 3a and b?
: The authors thank the referee for the comment. We apologize for confusing the reviewer. We did not count based on paraffin-embedded method. The ratio of capture, viability, and release was conducted before paraffin-embedded method. In other words, we measured the ratio based on anti-EpCAM Antibody-Immobilized Hydrogel Core. We added to method 2.4. Cell capture, viability, and fluorescence intensity. Changed Figure 2a and 2b were conducted in suspension. Changed Figure 2c was conducted on the gel. The capture efficiency was defined by calculation : [(Total counts) - (counts of remained cell after capture)] / (Total counts) x 100 (%). The retrieval efficiency was defined by calcualtion : [(released cell counts) / (Total counts)] x 100 (%). In order to confirm capture efficiency using fluorescence intensity, we conducted to stain of immunofluorescence. We confirmed the results of arbitrary unit before and after cell capture on anti-EpCAM Antibody-Immobilized Hydrogel Core.
Review2.10 :
This reviewer cannot understand why the fluorescent intensity of cells increase after cell capture by using hydrogels. Authors discussed as "This result predicts fluorescent-labeled cancer cells are specifically reacted with anti-EpCAM antibody-immobilized onto hydrogel". Why the intensity increases after the reaction with antibodies?
: The authors thank the referee for the comment. We apologize for confusing the reviewer. We added to method 2.4. Cell capture, viability, and fluorescence intensity. In order to confirm capture efficiency using fluorescence intensity, we conducted to stain of immunofluorescence on anti-EpCAM Antibody-Immobilized Hydrogel Core before and after cell capture. The cells were stained with a staining solution composed of Alexa fluor 647 anti-human CD326 (1:100 diluted in PBS, Biolegend). We confirmed the results of arbitrary unit before and after cell capture on anti-EpCAM Antibody-Immobilized Hydrogel Core. The “Pre-capture” was “stain of immunofluorescence on anti-EpCAM Antibody-Immobilized Hydrogel Core, before cell capture”. The “Post-capture” was “stain of immunofluorescence on anti-EpCAM Antibody-Immobilized Hydrogel Core, after cell capture”. We confirmed from fluorescence intensity arbitrary unit using a microplate spectrophotometer (Thermo Varioskan LUX) before and after EpCAM positive cell (Huh-7) adhesion on anti-EpCAM Antibody-Immobilized Hydrogel Core.
Review2.11 : S.3.3: L237, Fig. 4 The method of cell retrieval should be described. In addition, the method of calculation for both capture and retrieval should be clearly described. For example, what corresponds to 100% at the cell capture? (All cells in suspension were captured to the beads surfaces?) What is the objective of measuring cell viability in Fig. 4.
: The authors thank the referee for the comment. We apologize for confusing the reviewer. We added to method 2.4. Cell capture, viability, and fluorescence intensity. The capture efficiency was defined by calculation : [(Total counts) - (counts of remained cell after capture)] / (Total counts) x 100 (%). The retrieval efficiency was defined by calcualtion : [(released cell counts) / (Total counts)] x 100 (%). For cell release, the isolated cell using anti-EpCAM Antibody-Immobilized Hydrogel Core was added to a 0.5M EDTA solution (Invitrogen, CA, USA). The samples were centrifuged once more at 1,500 rpm at 24 °C for 5 min. To clarify the method on anti-EpCAM Antibody-Immobilized Hydrogel Core is important for maintaining cell viability. In order to demonstrate that the cells are still viable even after exposed to the effect of chemical stress during the isolation process, we checked the viability of captured cancer cells, which are induced by maintained biological information and reflected original tumor. As showed in results 3.2, the cell capture efficiency of Huh-7 was 92.9 ± 0.9%. We have confirmed that 91.7 ± 10.4% of the isolated cancer cells were released from the hydrogel core, and 89.5 ± 3.5 % of them remained still alive.
Review2.12 : If cells are analyzed with H&E staining, cells will be fixed and embedded to paraffin as demonstrated in this manuscript and cell viability might be not fundamental point in this procedure. If the authors suppose collection of cancer cells after the acquisition from cell suspension, authors should discuss advantages of the method suggested in this study in comparison with other methods such as MACS.
: Thank you for raising this important point. The authors thank the referee for the comment. The viable CTC cell is important to maintain living cell, because CTCs are induced by maintained biological status, involved genetic information, and reflected original tumor. After maintaining viable cell as much as possible, we conducted method of the paraffin-embedded based on anti-EpCAM Antibody-Immobilized Hydrogel Core.
We added to method 2.4 and Introduction. For example, the cell separation method using the magnetic-activated cell sorting (MACS) method is also a method that can separate cells while reducing the contamination of cells. However, when using MACS, in order to separate various cells at the same time, it is necessary to use specific antibodies and magnetic beads together with magnetic separating tools. These materials are consumable materials and are also expensive, therefore, there is a disadvantage that more time and cost are consumed compared to our hydrogel-based methods.

Round 2
Reviewer 1 Report
The revised manuscript has become understandable.
Crituque is that: Describing the main evaluation item and accessory evaluation item is needed in the bottom of introduction.
Author Response
The revised manuscript has become understandable.
Crituque is that: Describing the main evaluation item and accessory evaluation item is needed in the bottom of introduction.
> We appreciate the reviewer for a favorable comment. At the bottom of the introduction section, we described the main and subsidiary evaluation items, respectively. We modified Figure 5a, in order to make it easier for subscribers to see. Once again, we thank the reviewer for the valuable comments.
> Thank you for your comment. Additionally, we added Funding and Acknowledgments : This work was supported by Bio & Medical Technology Development Program of the National Research Foundation (NRF) & funded by the Korean government (NRF-2019M3E5D1A02068557).

Reviewer 2 Report
Revisions were well performed. Therefore, this manuscript can be acceptable after the following minor revision.
(1) Numbers of cancer cells doped in blood samples should be described.
Author Response
Revisions were well performed. Therefore, this manuscript can be acceptable after the following minor revision.
(1) Numbers of cancer cells doped in blood samples should be described.
> We appreciate the reviewer for a favorable comment. We revised the manuscript for addition of Numbers of cancer cells doped in blood samples. We modified Figure 5a, in order to make it easier for subscribers to see. Once again, we thank the reviewer for the valuable comments.
> Thank you for your comment. Additionally, we added Funding and Acknowledgments : This work was supported by Bio & Medical Technology Development Program of the National Research Foundation (NRF) & funded by the Korean government (NRF-2019M3E5D1A02068557).
